# Low-Dose-Rate Irradiation Suppresses the Expression of Cell Cycle-Related Genes, Resulting in Modification of Sensitivity to Anti-Cancer Drugs

**DOI:** 10.3390/cells11030501

**Published:** 2022-01-31

**Authors:** Kiichi Shimabukuro, Takahiro Fukazawa, Akinori Kanai, Hidehiko Kawai, Kengo Mekata, Nobuyuki Hirohashi, Naoya Kakimoto, Keiji Tanimoto

**Affiliations:** 1Department of Radiation Disaster Medicine, Research Institute for Radiation Biology and Medicine, Hiroshima University, Hiroshima 734-8553, Japan; d186571@hiroshima-u.ac.jp (K.S.); m202918@hiroshima-u.ac.jp (K.M.); hirohasi@hiroshima-u.ac.jp (N.H.); 2Department of Oral and Maxillofacial Radiology, Graduate School of Biomedical and Health Sciences, Hiroshima University, Hiroshima 734-8553, Japan; kakimoto-n@hiroshima-u.ac.jp; 3Natural Science Center for Basic Research and Development, Hiroshima University, Hiroshima 734-8553, Japan; tfuka@hiroshima-u.ac.jp; 4Department of Molecular Oncology, Research Institute for Radiation Biology and Medicine, Hiroshima University, Hiroshima 734-8553, Japan; akkanai@edu.k.u-tokyo.ac.jp; 5Laboratory of Systems Genomics, Department of Computational Biology and Medical Sciences, Graduate School of Frontier Sciences, The University of Tokyo, Chiba 277-8562, Japan; 6Department of Nucleic Acids Biochemistry, Graduate School of Biomedical and Health Sciences, Hiroshima University, Hiroshima 734-8553, Japan; kawaih@hiroshima-u.ac.jp

**Keywords:** low-dose rate irradiation, hypoxia, *AURKB*, *FOXM1*, paclitaxel

## Abstract

The biological effects of low-dose-rate (LDR) radiation exposure in nuclear power plant accidents and medical uses of ionizing radiation (IR), although being a social concern, remain unclear. In this study, we evaluated the effects of LDR-IR on global gene expression in human cells and aimed to clarify the mechanisms. RNA-seq analyses demonstrated that relatively low dose rates of IR modify gene expression levels in TIG-3 cells under normoxic conditions, but those effects were attenuated under hypoxia-mimicking conditions. Gene set enrichment analysis demonstrated that LDR-IR significantly decreased gene expression related to cell division, cell cycle, mitosis, and the Aurora kinase B and FOXM1 pathways. Quantitative RT-PCR confirmed the down-regulation of *AURKB* and *FOXM1* genes in TIG-3 cells with LDR-IR or hypoxia-mimicking treatments without any dose-rate effect. Knock-down experiments suggested that HIF-1α and HIF-2α, as well as DEC1, participated in down-regulation of *AURKB* and *FOXM1* under DFOM treatments, but to a lesser extent under LDR-IR treatment. FACS and microscopic analyses demonstrated that LDR-IR induced G0/G1 arrest and increased micronucleus or chromosome condensation. Finally, MTT assays demonstrated that LDR-IR decreased sensitivity to paclitaxel or barasertib in TIG-3 cells but not in A549 cells. In conclusion, LDR-IR modifies global gene expression and cell cycle control, resulting in a reduction of sensitivity to anti-cancer chemotherapy in non-cancer cells and thus a reduction in untoward effects (GA).

## 1. Introduction

After the nuclear power plant accidents in Chernobyl and Fukushima, the effects of low-dose-rate ionizing radiation (LDR-IR) on the human body have been a public concern. In addition, medical exposures from radiotherapy, particularly diagnostic radiology including computed tomography (CT) scanning, have been increasing year by year [1]. Many researchers have therefore been working to elucidate the effects of LDR-IR, but the specific mechanisms remain unclear [2,3,4]. Recently, it has been reported that the harmful effects caused by not only high-dose-rate irradiation (HDL-IR) but also LDR-IR increased chromosomal abnormalities that might be related to carcinogenesis [5]. On the other hand, some beneficial effects of LDR-IR have been reported, including anti-aging, biological defense activation, and anti-cancer effects [6,7,8]. These effects, either harmful or beneficial, can vary greatly among various cell types and are unpredictable. In general, there are several known factors that affect cellular responses to ionizing radiation; these include oxygen tension, temperature, histone configuration, and dose rate. In fact, it is well-known that a shortage of oxygen (hypoxia) plays an important role in enhancing resistance to radiation treatments [9]. We also demonstrated that hypoxic signal attenuated IR-induced DNA damage responses, including apoptosis [10]. Under hypoxia, the hypoxia-inducible transcription factors regulate gene expression related to adaptation to hypoxia, such as angiogenesis, metabolic reprograming, regulation of apoptosis, cellular differentiation, and DNA damage responses [11,12,13]. It has been further demonstrated that a hypoxic microenvironment and an activated hypoxic signal in various cancer cells conferred resistance to chemo-radiotherapies [14,15,16]. On the other hand, the effects of hypoxia on cellular responses to LDR-IR have not been elucidated. Therefore, in this study, we evaluated cellular responses to LDR-IR and the effects of hypoxia on these responses.

## 2. Materials and Methods

### 2.1. Chemicals

All chemicals were analytical grade and were purchased from FUJIFILM Wako Pure Chemicals (Osaka, Japan), Sigma-Aldrich (St. Louis, MO, USA), Selleckchem (Huston, TX, USA), or Bristol Myers Squibb (New York, NY, USA).

### 2.2. Cell Culture

Human lung fibroblast cells (TIG-3), lung adenocarcinoma cells (A549), and hepatoblastoma cells (HepG2) were purchased from The Japanese Cancer Research Resource Bank. Human breast cancer cells (MCF-7 and MDA-MB-231) were purchased from the American Type Culture Collection. Cells were maintained in Eagle’s Minimum Essential Medium (MEM) or RPMI1640 (NACALAI TESQUE, Inc., Kyoto, Japan) containing 10% fetal bovine serum (FBS; BioWhittaker, Verviers, Belgium) and 100 µg/mL of kanamycin sulfate solution (FUJIFILM Wako Pure Chemical Corporation, Osaka, Japan). To mimic hypoxia, cells were treated with 10 μM of DFOM, producing 1% O_2_. The extent of DFOM treatment was set to that which activated HIF-1 and induced a target gene, *CA9*, in each cell type, equivalent to that under hypoxic conditions (1% O_2_ for 24 h) (Appendix A).

### 2.3. Irradiation Procedure

LDR-IR was conducted with 100, 500, or 1000 mGy/day (24 h) with a ^137^Cs γ-ray source device (Chugai Technos Corporation, Hiroshima, Japan) for 1−7 days in each experiment. Each dose was set by adjusting the distance of the cell culture incubator (37 °C, 5% CO_2_) from the radiation source. Instantaneous irradiation was performed at 800 mGy/min with a Gammacell^®^ 40 Exactor (Nordion International, Inc., Ottawa, ON, Canada) for 23−225 s.

### 2.4. Cell Proliferation and Drug Sensitivity Analyses

Cell proliferation capacity was evaluated with the MTT assay or by counting cell numbers. Cells were seeded on a 96-well plate and cultured for 1 day, then treated with DFOM or IR for the indicated periods. For determining the IC_50_ values of paclitaxel (Bristol Myers Squibb) or barasertib (Selleckchem), each treatment was performed for 3 days. After incubation under the experimental treatment, the MTT (3-(4,5-dimethylthial-2-yl)-2,5-diphenyltetrazalium bromide) formazan precipitate was dissolved in DMSO, and the absorbance at 570 and 650 nm (reference) was measured with an EMax^®^ Endopoint ELISA Microplate Reader (Molecular Devices LLC, San Jose, CA, USA). Cell numbers were counted with an IN Cell Analyzer 2000 (GE Healthcare, Arlington Heights, IL, USA) after DAPI staining. All experiments were replicated at least 3 times under similar conditions.

### 2.5. RNA-Seq Analysis

Total RNA was prepared from frozen cell pellets by using NucleoSpin^®^ RNA (MACHEREY-NAGEL GmbH&Co. KG, Düren, Germany), according to the manufacturer’s instructions, and analyzed with a Bioanalyzer (Agilent Technologies, Santa Clara, CA, USA). Deep sequencing was performed by using a SureSelect Strand Specific RNA Library Prep Kit (Agilent Technologies) and HiSeq2500 (Illumina, San Diego, CA, USA) with 51 bp single-end reads at the core facility of Hiroshima University. Sequenced reads were mapped to the human genome assembly hg19 using CASAVA 1.8.2 (Illumina, RRID:SCR_001802), and read counts were normalized as reads per kilobase of exon per million mapped (RPKM). To define up-regulated or down-regulated genes, the data were trimmed by removing genes whose RPKM values were less than 2, and the relative expression between non-irradiated and irradiated cells was estimated. Genes with more than 2.0- or less than 0.5-fold expression—compared with non-irradiated cells—were identified for each differentially expressed gene. Gene set enrichment analysis was then performed on the identified genes by using Metascape (https://metascape.org/gp/index.html#/main/step1, accessed on 24 November 2021). Overlapping genes were extracted by using jvenn, a plug-in for the jQuery javascript library (http://jvenn.toulouse.inra.fr/app/index.html, accessed on 24 November 2021). These RNA sequence data have been deposited in the DNA Data Bank of Japan’s Sequence Read Archive (https://www.ddbj.nig.ac.jp/dra/index-e.html; Accession No: DRA012887, accessed on 18 October 2021).

### 2.6. Quantitative Reverse Transcription–Polymerase Chain Reaction (RT-PCR) Analysis

One µg of total RNA extracted from each sample was reverse-transcribed using a High-Capacity cDNA Archive^TM^ Kit (Applied Biosystems, Waltham, MA, USA). A two-hundredth aliquot of cDNA was subjected to quantitative RT-PCR with primers (final concentration 200 nM each) and MGB probe sets (final concentration 100 nM; the Universal Probe Library [UPL], Roche Diagnostics, Basel, Switzerland) as shown in Appendix A for *AURKB*, *FOXM1*, *HIF1A*, *EPAS1* (*HIF2A*), *DEC1* (*BHLHE40*), *DEC2* (*BHLHE41*), *BAX*, and *BCL2*. The pre-developed TaqMan assay reagent (Applied Biosystems) for was used for *ACTB* (4326315E, Applied Biosystems) as an internal control. PCR reactions were carried out with a 7500 real-time PCR system (Applied Biosystems) under standard conditions. Gene expression levels were standardized by using pooled cDNA derived from 17 non-identical cancer cell lines, and relative expression was calculated by using *ACTB* expression as the denominator for each cell line.

### 2.7. Immunoblotting Analysis

To analyze protein expression, whole cell extracts were prepared from cultured cells as previously described [13]. Fifty µg of extracts was blotted onto nitrocellulose filters following SDS-polyacrylamide gel electrophoresis. Anti-Aurora kinase B (#3094, Cell Signaling Technology, Danvers, MA, USA) or anti-β-actin (A5441, Sigma-Aldrich) were used as primary antibodies, diluted 1:500. A 1:4000 dilution of anti-rabbit IgG or anti-mouse IgG horseradish peroxidase conjugate (#7076, #7074, Cell Signaling Technology) was used as a secondary antibody. Immunocomplexes were visualized by using the enhanced chemiluminescence reagent SuperSignal West Pico PLUS (Thermo Fisher Scientific, Waltham, MA, USA).

### 2.8. Knock-Down Analysis

For knock-down analysis, TIG-3 or A549 cells were transfected with siRNA specific for *HIF1A* (siHIF1A, SI02664053), *EPAS1* (siEPAS1, SI02663038), *DEC1* (siDEC1, SI00311976), or *DEC2* (siDEC2, SI00312004), or with non-specific siRNA (siNS, No. 1027310) (QIAGEN, Inc., Valencia, CA, USA) by using Lipofectamine^TM^ RNAiMAX (Thermo Fisher Scientific) for 24 h. Cells were harvested and stored at −80 °C until use.

### 2.9. Plasmid Constructs and Luciferase Reporter Analysis

Promoter regions of *AURKB* (1571 bp, chr17: 8,210,202–8,211,772) and *FOXM1* (1145 bp, chr12: 2,877,127–2,878,271) were amplified by PCR from HepG2 genomic DNA and subcloned into the *Nhe* I and *Xho* I sites of a luciferase reporter plasmid, pGL4.26 (Promega Corporation, Madison, WI, USA). The constructs were confirmed by sequence analysis with a Big Dye Terminator Cycle Sequencing kit and an ABI 3130 Genetic Analyzer (Applied Biosystems). Details of the expression plasmid vectors of pcDNA-FLAG, pcDNA-HIF-1α, pcDNA-HIF-2α, p3×FLAG-CMV-DEC1, or p3×FLAG-CMV-DEC2 were described previously [17,18].

TIG-3 cells were seeded into 24-well plates and cultured for 24 h. The pGL4.26-AURKB or pGL4.26-FOXM1 promoter reporter constructs (0.2 µg per well of a 24-well plate) were co-transfected with pcDNA-FLAG, pcDNA-HIF-1α, pcDNA-HIF-2α, p3×FLAG-CMV-DEC1, or p3×FLAG-CMV-DEC2 (0.001–0.1 µg per well), using 0.8 µL of TransIT-LT1 Transfection Reagent (Mirus Bio LLC, Madison, WI, USA). A Renilla luciferase vector (pRL-SV40, 1.0 ng per 15-mm well) (Promega Corporation) was used as a transfection efficacy control. Cells were incubated under normoxic, DFOM-treated, or irradiated (1000 mGy/day) conditions for 24 h after transfection, prior to analysis of luciferase reporter activity. Luciferase luminescence was measured by using Lumat LB9507 (Berthold Technologies, Bad Wildbad, Germany).

### 2.10. Cell Cycle Analysis

The cells collected after LDR-IR or DFOM treatment, as noted above, were immediately fixed with 70% ethanol and stained with propidium iodide (PI/RNase Staining Buffer; BD Biosciences, Franklin Lakes, NJ, USA) for 15 min at room temperature in the dark. FACS analysis was then performed with LSRFortessaX-20™ (BD Biosciences) and BD FACSDiva ™ software.

### 2.11. Immunofluorescence Analysis

TIG-3 or A549 cells grown on cover slips were treated with LDR-IR (100, 500, or 1000 mGy/day) or DFOM for 3 days. After incubation, cell nuclei were stained with 4–6-diamidino-2-phenylindole (DAPI). The subcellular distribution of fluorescence was observed and photographed with a BZ-8000 microscope (KEYENCE, Osaka, Japan). Typical images per field were selected and counted within each photograph.

### 2.12. Statistical Analysis

All of the statistical tests were performed with EZ-R version 1.54. The Tukey–Kramer method or *t*-tests were used to determine the *p*-value for post-hoc pairwise comparisons [19] when an ANOVA test revealed significant heterogeneity.

## 3. Results

### 3.1. Effects of LDR-IR on Cell Proliferation

At first, we compared sensitivity to γ-ray LDR-IR in several cell lines under normoxic conditions and found that lung fibroblast cells (TIG-3) were the most sensitive among them (Appendix A). We thus evaluated the cell proliferation capacities of TIG-3 and lung adenocarcinoma (A549) cells irradiated with LDR-IR at various dose rates (100, 500, or 1000 mGy/day) under normoxic or hypoxia-mimicking (DFOM) conditions (Figure 1). The cell proliferation capacity of TIG-3 cells was significantly suppressed with 1000 mGy/day of LDR-IR under normoxic conditions. That of TIG-3 cells was also suppressed with 500 mGy/day of LDR-IR, although not significantly. On the other hand, that of A549 cells was not suppressed with LDR-IR. DFOM treatment clearly suppressed the proliferation capacities of both cell types used here and therefore precluded the effects of LDR-IR that were observed under normoxic conditions (Figure 1).

### 3.2. Effects of LDR-IR on Global Gene Expression

We next evaluated the effects of the LDR-IR and DFOM treatments on global gene expression in TIG-3 cells by RNA-seq analysis. The number of genes for which expression increased to more than two-fold or decreased to less than 0.5-fold were identified at all IR dose ranges under normoxic conditions, and the number of genes with such fold changes increased in a dose-dependent manner (expression increased in 16 genes and decreased in 15 genes with 100 mGy/day, increased in 22 and decreased in 51 with 500 mGy/day, and increased in 134 and decreased in 372 with 1000 mGy/day). DFOM treatment attenuated these responses to LDR-IR (Figure 2a). Venn diagrams indicated that some of the same genes were up-regulated at different doses, but there seemed to be a change in down-regulated genes above 500 mGy/day (Appendix A). Venn diagrams also indicated that few of the same genes were also regulated after 1000 mGy/day IR under normoxic conditions and DFOM treatment (Figure 2b,d). Gene set enrichment analysis showed that 132 genes that were specifically up-regulated after 1000 mGy/day IR under normoxic conditions were mostly those related to functions that occur in the extracellular matrix (Figure 2c). On the other hand, 372 down-regulated genes were mostly related to cell division and cell cycle control through the Aurora kinase B and FOXM1 pathways (Figure 2e).

### 3.3. Effects of LDR-IR on Expression of AURKB and FOXM1

Since the gene set enrichment analysis suggested that LDR-IR suppressed cell division and affected cell cycle control through the Aurora kinase B and FOXM1 pathways, respectively, we tried to evaluate the effects of LDR-IR and DFOM treatment on the expression of *AURKB* and *FOXM1*. Quantitative RT-PCR confirmed the results of RNA-seq, i.e., that the expression levels of *AURKB* and *FOXM1* decreased significantly with LDR-IR in TIG-3 cells in a dose-dependent manner (Figure 3a). These expression levels also decreased significantly with DFOM treatment. Although it was not significant, they decreased further in TIG-3 cells with LDR-IR under DFOM, suggesting that different mechanisms might be involved in LDR-IR and DFOM treatment. The expression of *AURKB* also decreased in LDR-irradiated and DFOM-treated A549 cells, but the expression of *FOXM1* decreased only moderately under these treatments (Figure 3a). The inhibitory effects of DFOM treatments were observed in a time-dependent manner, and the highest appeared at 72 h in TIG-3 and A549 cells (Figure 3b). Similar effects of IR on the expression of *AURKB* and *FOXM1* were observed in cells instantaneously irradiated with identical total doses (Figure 3c). Down-regulation of Aurora kinase B was also confirmed by an assessment of the protein levels (Figure 3d).

### 3.4. Mechanisms of Altered AURKB and FOXM1 Expression

To clarify the mechanisms by which regulation of *AURKB* and *FOXM1* gene expression was affected by LDR-IR or DFOM treatment, experiments were performed with knock-down of hypoxia-regulated transcription factor genes. The expression of *HIF1A* (HIF-1α), *EPAS1* (HIF-2α), *DEC1*, or *DEC2* genes in TIG-3 cells with LDR-IR or DFOM treatment decreased significantly after transient transfection with specific siRNA (Figure 4a,c). The expression of *AURKB* decreased significantly in *HIF1A*, *EPAS1*, and *DEC1* knock-down cells, and DFOM-induced down-regulation of *AURKB* was suppressed in these cells (Figure 4b). The expression of *FOXM1* also decreased significantly in *HIF1A*, *EPAS1*, and *DEC1* knock-down cells, and DFOM-induced down-regulation of *FOXM1* was suppressed in these cells; in fact, it was up-regulated in *DEC1* knock-down cells. These results suggest the involvement of HIF-1α, HIF-2α, and DEC1 in the mechanisms of DFOM-induced down-regulation. On the other hand, IR-induced down-regulation of *AURKB* and *FOXM1* continued to be observed in *HIF1A*, *EPAS1*, *DEC1*, and *DEC2* knock-down cells, although the decreased expression was not significant in some situations (Figure 4d). The expression of *AURKB* and *FOXM1* in *DEC2* knock-down cells increased slightly in both control and irradiated cells, suggesting some involvement of DEC2 in the mechanisms of IR-induced down-regulation. Similar results were observed in A549 cells (Appendix A). The luciferase reporter assay with subcloned *AURKB* and *FOXM1* gene promoters further indicated HIF-2α as a candidate activator of these genes in TIG-3 cells, but no other modification was observed under the DFOM or LDR-IR treatments, or with transient co-transfections (Appendix A).

### 3.5. Effects of LDR-IR on the Cell Cycle and Nuclear Morphology

Since RNA-seq and subsequent gene set enrichment analysis suggested that LDR-IR suppresses cell division and cell cycle control, we evaluated the effects of LDR-IR on cell cycle regulation. Flow cytometry with PI-stained TIG-3 cells demonstrated that 1000 mGy/day IR significantly increased the proportion of cells in the G0–G1 phase and more than 500 mGy/day IR significantly decreased the proportion of cells in the S phase (Figure 5a). A549 cells also showed a significant increase in the proportion of cells in the G0–G1 phase after LDR-IR, and with more than 500 mGy/day IR, the proportion in the S and G2/M phases decreased significantly (Figure 5a). Under DFOM treatment, the number of cells in the G0–G1 phase decreased and the number in the S phase increased in both TIG-3 and A549 cells, with the changes being strongly evident. DFOM treatment also led to a significant increase in the number of TIG-3 cells in the G2/M phases but a decrease in the number of A549 cells in the G2/M phases. Microscopic observations with DAPI nuclear staining further demonstrated that the frequency of TIG-3 cells in the M phase decreased significantly with LDR-IR or DFOM treatment, whereas the number of A549 cells in the M phase showed no remarkable changes with LDR-IR and a slight decrease with DFOM treatment (Figure 5b and Appendix A). Increased numbers of micronuclei and multiple nuclei, indicating dead cells, were also observed (Figure 5c). The number of micronuclei was significantly increased with LDR-IR in TIG-3 cells. A similar result was observed in A549 cells, although it was not significant. The number of dead TIG-3 cells also increased with LDR-IR, but LDR-IR did not lead to a change in the number of dead A549 cells. On the contrary, with DFOM treatment, the number of dead A549 cells increased significantly, but there was a change in the number of dead TIG-3 cells. Quantitative RT-PCR indicated that *BCL2* expression decreased significantly after LDR-IR in TIG-3 but not in A549 cells, resulting in reduced anti-apoptotic signals only in TIG-3 cells (Figure 5d). On the other hand, no remarkable change was observed in the expression level of *BAX* in either cell type.

### 3.6. Effect of LDR-IR on Sensitivity to Cell Cycle-Targeting Anti-Cancer Drugs

Paclitaxel is in the taxane family of anti-cancer drugs and works by interfering with the function of microtubules during cell division, resulting in its efficacy as a cell cycle-targeting drug [20,21]. Since LDR-IR modified the cell cycle distribution in TIG-3 cells, we evaluated the effect of LDR-IR on cell sensitivity to paclitaxel. MTT assays interestingly demonstrated that pre-treatment with LDR-IR before paclitaxel led to significantly reduced sensitivity to paclitaxel in TIG-3 cells only (Figure 6a). MTT assays also demonstrated reduced sensitivity to the Aurora kinase B specific inhibitor, barasertib, in TIG-3 but not in A549 cells (Figure 6a). Pre-treatment with DFOM appeared to reduce sensitivity to paclitaxel or barasertib in both TIG-3 and A549 cells (Figure 6b). Importantly, another cancer cell, HSC-2, also showed unaltered sensitivity to paclitaxel and barasertib with LDR-IR, while the sensitivity of these cells was reduced with DFOM (Appendix A).

## 4. Discussion

The biological effects of low-dose-rate (LDR) radiation exposure in medical irradiation (IR) have been a source of public concern for a long time, and such concern was highlighted after the nuclear power plant accidents in Chernobyl and Fukushima. Much effort has been thus been spent on conducting studies to elucidate the mechanistic effects of LDR-IR, but much remains unclear [2,3,4]. In this study, we were able to elucidate some effects of LDR-IR on global gene expression and cell cycle control, resulting in useful modification of sensitivity to anti-cancer chemotherapy in a cell-specific manner.

We first evaluated the effects of relatively low-dose-rate IR on the proliferation capacity of several human cell lines, and found that lung fibroblast TIG-3 cells exhibited the greatest suppression with 500–1000 mGy/day of LDR-IR among the cells tested (Figure 1 and Appendix A). On the other hand, lung cancer A549 cells did not exhibit such suppression with LDR-IR (Figure 1), and other cancer cell lines showed smaller effects of growth suppression than that in TIG-3 cells (Appendix A). These results suggest that a subtle IR dose rate produces cell-type-specific effects, although a larger number of cell lines should be tested. The limitation of this study is that only a few cell lines were analyzed in detail; however, that LDR-IR leads to differential regulation of proliferative capacity between normal and cancer cells was also supported by a previous study [22]. We also compared the effect of treatment with a hypoxia-mimicking reagent, DFOM, on cell proliferation but did not find any difference, suggesting that the hypoxia-mimicking reagent produced similar effects in those cells. RNA-seq analyses further demonstrated that gene expression levels were modified in TIG-3 cells after LDR-IR under normoxic conditions, in a dose-dependent manner, but those effects were strongly attenuated under hypoxia-mimicking conditions. Venn diagrams indicated that some genes were commonly up-regulated at different doses, but there seemed to be a change after 500 mGy/day in down-regulated genes, in that few were common to widely different doses (100 and 1000 mGy; Appendix A). These results interestingly suggest that further fine-tuning of the dose rate of IR might be necessary to facilitate clarifying the detailed effects on global gene expression.

We then performed gene set enrichment analysis to study the mechanisms of regulation of gene expression involved in the LDR-IR- and hypoxia-related effects. Specifically, up-regulated genes in irradiated cells mostly involved genes related to extracellular matrix-related functions, such as response to wounding, connective tissue development, muscle cell differentiation, regulation of epithelial cell proliferation, and collagen metabolic processes (Figure 2c). These results suggest a relationship between modified gene expression and some beneficial effects of LDR-IR, including anti-aging, biological defense activation, and anti-cancer effects, which were reported previously [6,7,8]. On the other hand, LDR-IR was associated with a decrease in the expression of genes related to cell division, cell cycle, mitosis, and the AURORA B and FOXM1 pathways (Figure 2e). Quantitative RT-PCR further confirmed the down-regulation of *AURKB* and *FOXM1* genes in TIG-3 cells after LDR-IR. When TIG-3 cells were exposed to the same doses of radiation instantaneously, a similar down-regulation of *AURKB* and *FOXM1* genes was observed, suggesting that there is no dose-rate effect. Since treatment with the hypoxia-mimicking reagent DFOM attenuated the down-regulation of those genes after LDR-IR, we speculate that hypoxic signal may overcome responses to LDR-IR. In this study, LDR-IR affected *AURKB* and *FOXM1* expression similarly in TIG-3 and A549 cells, but LDR-IR affected cell proliferation differently in these cell types (Figure 1 and Figure 3a). It is known that ATM-mediated p53 plays an important role in the response to radiation [23]. To compare the p53 signal pathway between TIG-3 and A549 cells, a public database indicates that more mutations in p53-related genes were found in A549 cells, suggesting an alteration in p53 signal (Appendix A). Further analysis of p53 signal in TIG-3 and A549 cells with LDR-IR would therefore be interesting.

The contribution of well-known hypoxia-inducible transcription factors—HIF-1α, HIF-2α, DEC1, and DEC2—to the down-regulation of *AURKB* and *FOXM1* genes with LDR-IR as well as DFOM treatment was evaluated by using specific siRNA; we found that HIF-1α and HIF-2α, as well as DEC1, might participate in the down-regulation of *AURKB* and *FOXM1* with DFOM treatment, but not to a great extent with LDR-IR (Figure 3c). In *DEC2* knock-down cells, expression of *AURKB* and *FOXM1* was slightly increased in both control and irradiated cells, suggesting some involvement of DEC2 in IR-induced down-regulation mechanisms. Promoter reporter experiments further suggested that the down-regulation of *AURKB* and *FOXM1* gene expression after LDR-IR might not be due to transcriptional regulation, but rather to other mechanisms such as regulation of RNA stability or epigenetic regulation. There are, in fact, some reports about the relevance of promoter regulation and the N^6^-methyladenosine of mRNA [24,25,26,27]. Further analyses are necessary to clarify the detailed mechanisms.

Since RNA-seq and subsequent gene set enrichment analysis suggested that LDR-IR suppressed cell division and cell cycle control, we also evaluated the effects of LDR-IR on cell cycle regulation. After LDR-IR, the G0–G1 population size increased, whereas the S and G2/M population sizes decreased, in both TIG-3 and A549 cells, suggesting activation of the G1/S checkpoint (Figure 5a). Microscopic observations, however, indicated decreased mitosis in TIG-3 but not A549 cells, suggesting that regulation of the G2/M transition under LDR-IR differs between those two cell types. DFOM treatment strongly decreased the G0–G1 population size and increased the S population size in both TIG-3 and A549 cells. DFOM interestingly increased the G2/M population size in TIG-3 cells but decreased it in A549 cells, also suggesting differential regulation of cell cycle checkpoints: activation of the M phase checkpoint in TIG-3 cells but the S phase checkpoint in A549 cells. These results suggest that LDR-IR and hypoxic signaling regulate the cell cycle in a cell-type-specific manner. The majority of double-strand breaks (DSBs) are repaired through non-homologous end-joining (NHEJ) in somatic cells. However, homologous recombination repair (HRR) is known to be enhanced in embryonic cells [28], in which it is invoked predominantly during the S and G2 phases [29,30]. NHEJ is also reported to act in the G1 and early S phases in vertebrate cells [31,32]. In this study, regulation of the G2/M transition by LDR-IR was suggested to differ between TIG-3 and A549 cells. Since the TIG-3 cell line is derived from embryos, LDR-IR might predominantly affect HRR, resulting in inhibition of the G2/M transition.

Microscopic observations indicated increased micronuclei in LDR-irradiated TIG-3 cells (Figure 5c). LDR-IR also significantly increased the number of dead TIG-3 cells but not of A549 cells, although the proportions of dead cells were small (Figure 5c). That *BCL2* expression was down-regulated by LDR-IR only in TIG-3 cells further supports these observations (Figure 5d). These results suggest that LDR-IR actually induces DNA damage and subsequent cell cycle arrest, resulting in suppressed proliferation of TIG-3 cells. Since we observed the cells for only 3 days after IR in this study, the long-term effects could not be evaluated, so longer observations are desired.

Finally, we investigated whether LDR-IR affected the sensitivity of anti-cancer drugs targeting the cell cycle or Aurora kinase B, and interestingly found that pre-treatment with LDR-IR significantly reduced the sensitivity to paclitaxel or barasertib in TIG-3 but not in A549 cells. On the other hand, DFOM pre-treatment reduced sensitivity to both drugs in both cells (Figure 6b). The molecular mechanisms of these results remain unclear, but one possible explanation is that LDR-IR significantly decreased the M phase of TIG-3 but not A549 cells. Furthermore, DFOM treatment decreased the number of cells in the M phase in both cell types. Such differential regulation of the cell cycle might allow for the differential control of sensitivity to paclitaxel, allowing its cytotoxic effects to be better focused on cancer cells and less damaging to healthy cells. Some reports suggested that the *AURKB* gene is involved in the mechanisms of susceptibility to paclitaxel for NSCLC, susceptibility to cetuximab for head and neck SCC, and susceptibility to vemurafenib for melanoma, and that *FOXM1* is involved in susceptibility to paclitaxel or herceptin for breast cancer [33,34,35,36]. Taken together, our results demonstrate that LDR-IR modifies chemo-sensitivity through the regulation of genes related to these anti-cancer-treatment mechanisms.

In summary, we clarified the effects of LDR-IR on cell cycle control through the cell-type-specific regulation of *AURKB* and *FOXM1* gene expression. On that basis, we were able to suggest a promising chemo-radiation protocol combining LDR-IR and cell cycle-targeting anti-cancer drugs. Further investigation is necessary to clarify the detailed mechanisms and to develop effective radiological protection or anti-cancer treatments.

## Figures and Tables

**Figure 1 cells-11-00501-f001:**
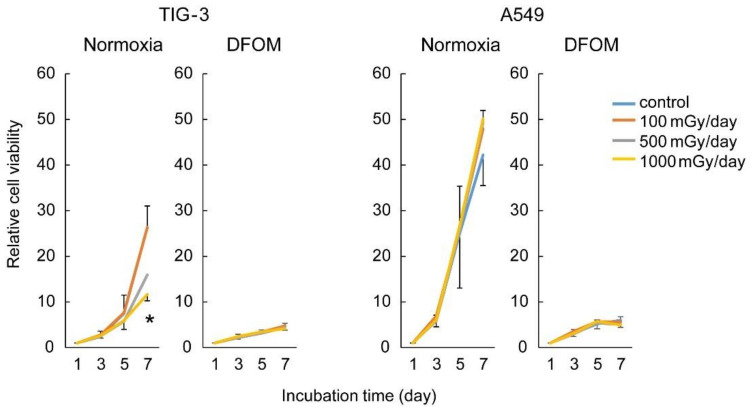
Effects of LDR-IR on cell proliferation. The relative cell viability of TIG-3 and A549 cells against LDR-IR under normoxic or hypoxia-mimicking (DFOM treatment) conditions was evaluated with the MTT assay. LDR-IR or DFOM treatment was started on Day 1, and the MTT assay was performed on Days 1, 3, 5, and 7. Values are means and SD (*n* = 3); * *p* < 0.05 vs. control.

**Figure 2 cells-11-00501-f002:**
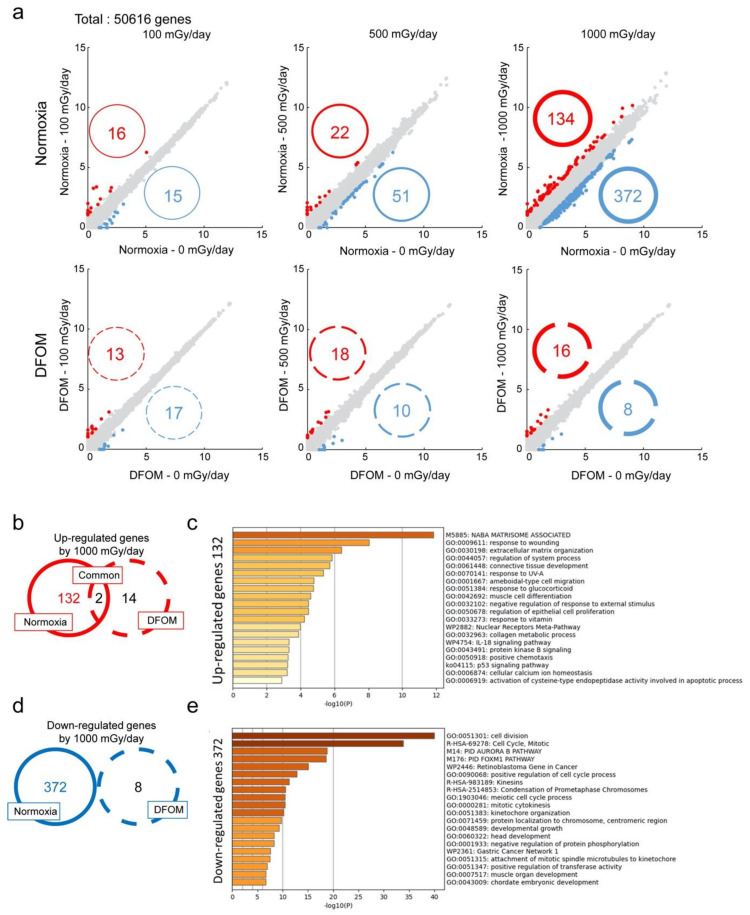
Effects of LDR-IR on global gene expression. (**a**) Gene expression levels in TIG-3 cells without or with LDR-IR under normoxic or DFOM-treated conditions were evaluated by RNA-seq. The upper three panels show the results under normoxic conditions with expression at 0 mGy/day on the *x*-axis and expression after LDR-IR at different dose rates on the *y*-axis; the lower panels show analogous results under DFOM treatment. Red indicates up-regulation, blue indicates down-regulation, and gray indicates unchanged genes (between 0.5- and 2.0-fold). Numbers in circles indicate the number of genes with modified expression levels. (**b**,**d**) Venn diagrams showing up-regulated (**b**) or down-regulated (**d**) genes after 1000 mGy/day IR under normoxia (left circle) or under DFOM treatment (right circle). (**c**,**e**) Gene set enrichment analyses were performed with Metascape. Up-regulated (**c**) or down-regulated (**e**) genes in TIG-3 cells at 1000 mGy/day under normoxic conditions were enriched by gene ontology (GO) terms.

**Figure 3 cells-11-00501-f003:**
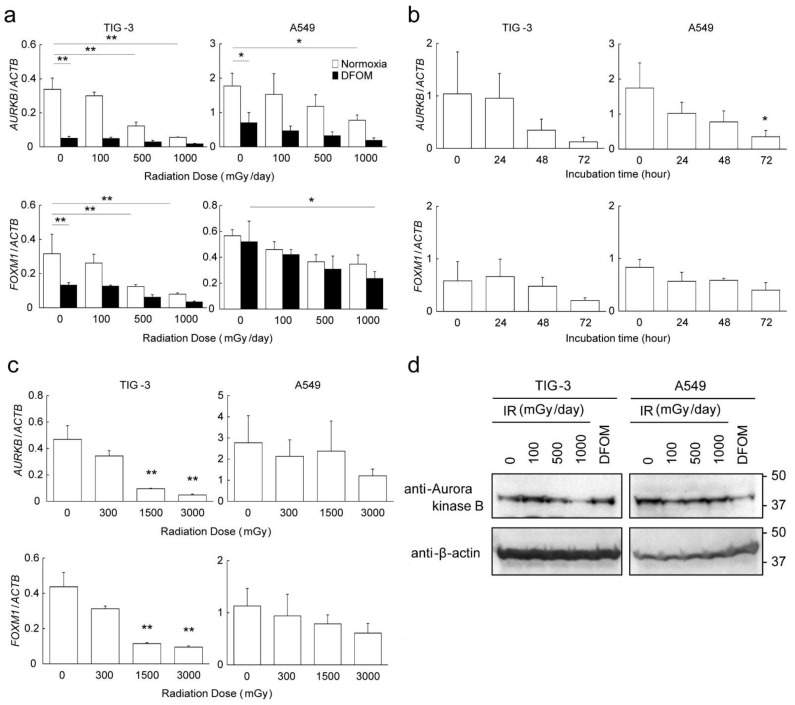
Effects of LDR-IR on the expression of *AURKB* and *FOXM1*. Expression levels of *AURKB* and *FOXM1* were evaluated by quantitative RT-PCR. (**a**) TIG-3 and A549 cells were irradiated with 100, 500, or 1000 mGy of γ-rays per day for 3 days under normoxic or DFOM-treated conditions. (**b**) Cell lines were treated with DFOM for 24, 48, and 72 h. (**c**) Cell lines were irradiated with 300, 1500, or 3000 mGy of γ-rays, the same total doses as in the LDR-IR experiment above but with instantaneous exposure. (**a**–**c**): Relative gene expression levels were calculated as the ratio to *ACTB* expression. Values are represented as means and SD (*n* = 3). * *p* < 0.05; ** *p* < 0.01. (**d**) Protein levels of Aurora kinase B in LDR-irradiated or DFOM-treated TIG-3 and A549 cells were analyzed via immunoblotting. β-actin was used as an internal loading control. Representative images are shown from three independent experiments.

**Figure 4 cells-11-00501-f004:**
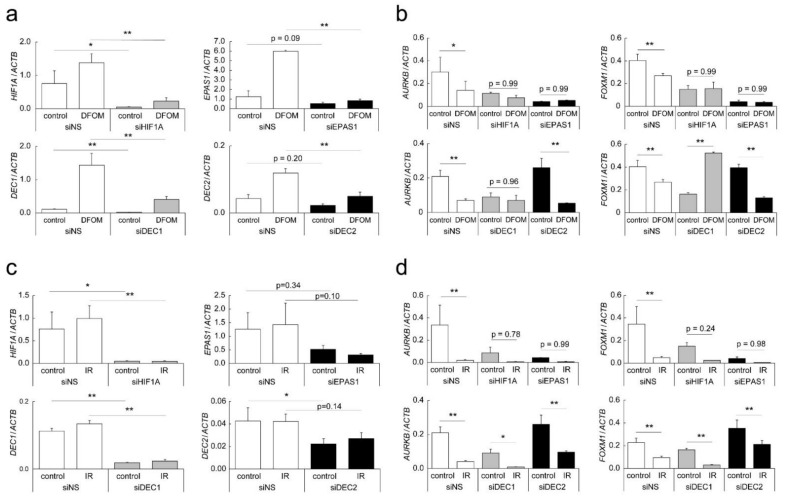
Regulatory mechanisms of *AURKB* and *FOXM1* expression. Knock-down experiments were performed by using transient transfection with specific siRNA for *HIF1A*, *EPAS1*, *DEC1*, and *DEC2*. (**a**,**c**) Expression of *HIF1A*, *EPAS1*, *DEC1*, and *DEC2* in DFOM-treated (**a**) or irradiated (1000 mGy/day) (**c**) TIG-3 cells analyzed by quantitative RT-PCR. (**b**,**d**) Expression of *AURKB* and *FOXM1* in DFOM-treated (**a**) or irradiated (1000 mGy/day) (**c**) TIG-3 cells analyzed by quantitative RT-PCR. (**a**–**d**) Relative gene expression levels calculated as the ratio to that of *ACTB*. Values are means and SD (*n* = 3). * *p* < 0.05; ** *p* < 0.01.

**Figure 5 cells-11-00501-f005:**
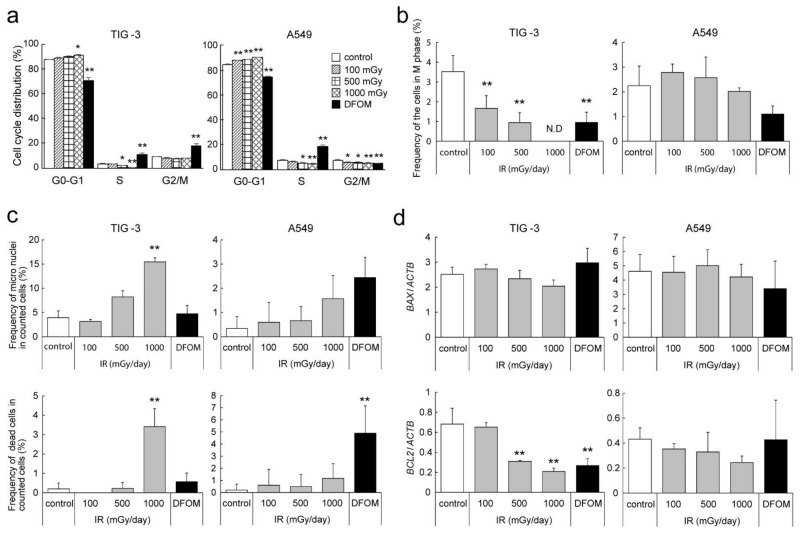
Effects of LDR-IR on cell cycle distribution and nuclear morphology. (**a**) Flow cytometry analysis was performed for evaluating the cell cycle in LDR-irradiated or DFOM-treated TIG-3 and A549 cells. (**b**,**c**) Microscopic observation was performed with DAPI nuclear staining. Frequencies of M phase cells (**b**), and micronuclei and dead cells (**c**). (**d**) Expression of *BAX* and *BCL2* was analyzed via quantitative RT-PCR. Relative expression level was calculated as the ratio to *ACTB* levels. (**a**–**c**) Values are means and SD (*n* = 3). * *p* < 0.05 vs. control; ** *p* < 0.01 vs. control.

**Figure 6 cells-11-00501-f006:**
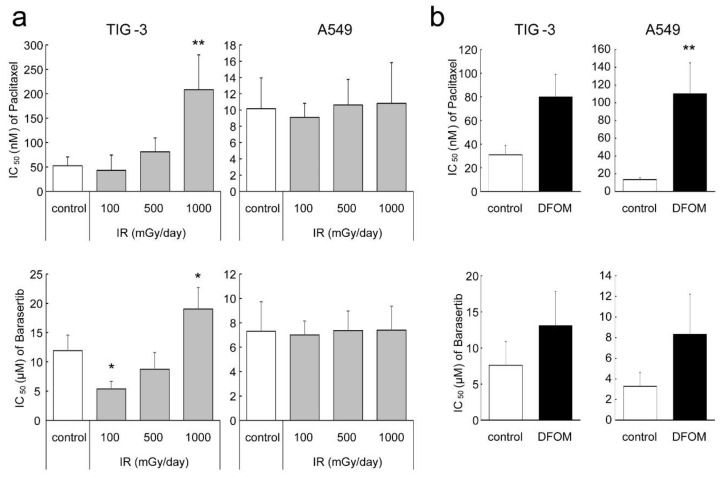
Effect of LDR-IR on sensitivity to cell cycle-targeting anti-cancer drugs. IC_50_ values in response to paclitaxel and barasertib in TIG-3 and A549 cells pre-treated with LDR-IR (*n* = 4) (**a**) or DFOM (*n* = 5) (**b**) were evaluated with the MTT assay. (**a**,**b**) Values are means and SD. * *p* < 0.05 vs. control; ** *p* < 0.01 vs. control.

## Data Availability

The data that support the findings of this study are openly available in the DNA Data Bank of Japan’s Sequence Read Archive (https://www.ddbj.nig.ac.jp/dra/index-e.html; Accession No: DRA012887, accessed on 18 October 2021).

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
