# Peer review of "Low-Dose-Rate Irradiation Suppresses the Expression of Cell Cycle-Related Genes, Resulting in Modification of Sensitivity to Anti-Cancer Drugs"

_cells, 2022, doi:10.3390/cells11030501_

Round 1

Reviewer 1 Report

This article has consistently and systematically proved the fact of an additive effect on tumor cells that hypoxia is a protective factor for cancer cells. The study of this fact was carried out very scrupulously and thoroughly. However, for the reviewer, the choice of the object remains unclear - why the embryonic line TIG-3 was chosen for the study. As a rule, embryonic cells are especially sensitive to hypoxia and their proteomes, which differ somewhat, for example, from the protein of normal fibroblasts. Moreover, the genome of fibroblasts is almost always much more tolerant to hypoxia associated with the peculiarities of metabolic processes and the functioning of connective tissue. In general, it would be much more correct to build a study on the comparison of several cell lines, compare the proteomes of normal fibroblasts, any embryonic cells, and take several adherent cultures from solid tumors. And only if in all these tissues the theory put forward by the authors of the lower damaging ability of ionizing radiation and chemotherapy is correct, then it has the right to exist. It seems to me that it is impossible to draw such conclusions based on the proteome of only two In the meantime, I recommend the editor to accept this article after the authors make minor additional edits.
So, in the introductory part, I recommend to more fully discuss and analyze the research carried out in this area by other authors, highlight the main achievements and existing gaps, and, on the basis of these data, formulate the objectives of this research.
In addition, in view of the above and based on the results of the research, I strongly recommend expanding the conclusions.
Also, once again re-read the experimental part and the description of the methods used to eliminate inaccuracies.
For example, when studying the cell cycle, it was indicated that 100% ethanol (???) was taken, usually 70% is used for these purposes.cell cultures.

Author Response

Comments of Reviewer #1

This article has consistently and systematically proved the fact of an additive effect on tumor cells that hypoxia is a protective factor for cancer cells. The study of this fact was carried out very scrupulously and thoroughly.

We appreciate the reviewer’s comments and careful reading of our manuscript.

1: However, for the reviewer, the choice of the object remains unclear - why the embryonic line TIG-3 was chosen for the study. As a rule, embryonic cells are especially sensitive to hypoxia and their proteomes, which differ somewhat, for example, from the protein of normal fibroblasts. Moreover, the genome of fibroblasts is almost always much more tolerant to hypoxia associated with the peculiarities of metabolic processes and the functioning of connective tissue. In general, it would be much more correct to build a study on the comparison of several cell lines, compare the proteomes of normal fibroblasts, any embryonic cells, and take several adherent cultures from solid tumors. And only if in all these tissues the theory put forward by the authors of the lower damaging ability of ionizing radiation and chemotherapy is correct, then it has the right to exist. It seems to me that it is impossible to draw such conclusions based on the proteome of only two. In the meantime, I recommend the editor to accept this article after the authors make minor additional edits.

So, in the introductory part, I recommend to more fully discuss and analyze the research carried out in this area by other authors, highlight the main achievements and existing gaps, and, on the basis of these data, formulate the objectives of this research.

In addition, in view of the above and based on the results of the research, I strongly recommend expanding the conclusions.

Response 1: That is an important point. We chose TIG-3 cells for RNA-seq analyses for several reasons: 1) we had just two lines of normal cells, TIG-3 and KD lip fibroblast, but KD did not grow well enough for use in the experiments; 2) we had to choose one cell line because of budget restrictions; and 3) we knew that TIG-3 could be a good model line for radiation experiments, because it is sensitive to irradiation and also to hypoxia as the reviewer pointed out. Based on the reviewer’s suggestions about the specificity of TIG-3 as a germ cell, we added sentences in the Discussion (lines 422-429). We focused on signals of the response to low-dose rate irradiation, and found interesting molecular responses, including down-regulation of cell cycle related gene expressions. We are planning to broaden the subject of our investigation in the future to be able to generalize our findings. There are few previous studies of in vitro low-dose rate irradiation, but we added a reference in which several cell lines were compared. We also added a sentence about our research limitation (Lines 361-365).

2: Also, once again re-read the experimental part and the description of the methods used to eliminate inaccuracies.

For example, when studying the cell cycle, it was indicated that 100% ethanol (???) was taken, usually 70% is used for these purposes.

Response 2: That is correct. We checked and corrected the Methods section, including the ethanol concentration as pointed out by the reviewer.

Reviewer 2 Report

In this paper Shimabukuro K. et al., show suppression of cell-cycle related genes after exposure to low-dose-rate radiation (LDRR) in normal (TIG-3) lung cells. Cancer cells (A549) are claimed to show different responses in comparison to normal (TIG-3) cells.

The subject of this study is relevant which may help to access the effects of LDRR in case of nuclear power plant accidents or during the medical uses of IR in diagnosis. The biggest limitation of results in the present form is that they are based on observations with only one set of cell lines and thus cannot be claimed to be valid in general for normal versus cancer cells as claimed by the authors in this paper.

The paper needs a major revision to clarify issues which are listed below as my comments to the authors.

Comments to the authors:

  1. Lanes 343-345: Authors mentioned that: We first evaluated effects of relatively low dose rate-IR on proliferation capacity of several human cell lines, and found that lung fibroblast TIG-3 cells exhibited suppression with 500-1000 mGy/day of LDR-IR.  Does this mean that out of several normal human cell lines which were tested, only TIG-3 showed the observed suppression following LDRR? If yes, then the claim of differential response in normal versus cancer cells after LDRR is not experimentally justified. Authors should include the data for all tested cell lines (both normal and cancer) and if only one cell line shows the effect among many normal cell lines tested then should avoid generalizing the findings. A549 did not show any changes in proliferation but what about the other cancer cell lines tested? Authors should clarify these issues and include the relevant data with all cell lines tested.
  2. The irradiation procedure is the key to this kind of experiment. The irradiation procedure should be presented as an independent section under Materials and Methods (M&M) and should be sufficiently elaborated to help the readers understand the exact procedure for irradiation. The authors should mention clearly which dose was delivered on the indicated dose rate? If the dose was 100, 500, or 1000mGy then does it mean that it took a whole day to irradiate the sample! Does per day mean 12 or 24 hours? Did some experiments take 1-7 days for irradiation (as mentioned in the M&M section)? If yes how the temperature and CO2 were maintained during irradiation?
  3. Figure 1: Why MTT assay was preferred over the clonogenic survival assay? MTT only shows the impact on short-term proliferation. The addition of clonogenic survival data is necessary to access the effect of LDRR properly.
  4. Figure 2: Why the gene expression was not checked in A549 cells! Although no effect of LDR-IR was observed on the proliferation of A549, still it is important and valid to compare gene expression analysis between normal vs cancer cells.
  5. Figure 5a: LDRR exerts similar effects on the cell cycle in TIG-3 and A549 cells which suggests that both normal and cancer cells have a similar impact of LDRR! This result contradicts the claim of the cell type-based response of LDRR.
  6. Figure 5b: It is surprising that TIG-3 and A549 show similar cell cycle responses in Figure 1A but clear differences on M-phase cells. Authors should check the mitotic cells using H3ps10 staining combined with cell cycle distribution to strengthen their results claiming differences in G2-M transition between these two cell lines. In addition, the representative images of mitotic cells observed under the microscope should be included.
  7. Figure 6: This is an important observation and this response should also be checked in several cell lines which authors already checked for proliferation during the screening phase (please also see comment 1). Inclusion of such data with clear differences among many normal and cancer cell lines would strengthen the findings significantly.
  8. The supplementary data was not accessible using the given link and showed the Error 404-File not found!

Author Response

Response to Reviewer #2

We appreciate the reviewer’s comments and suggestions in regard to inadequacies in our original draft, and we apologize for difficulties encountered by the reviewer.

1: Lanes 343-345: Authors mentioned that: We first evaluated effects of relatively low dose rate-IR on proliferation capacity of several human cell lines, and found that lung fibroblast TIG-3 cells exhibited suppression with 500-1000 mGy/day of LDR-IR.  Does this mean that out of several normal human cell lines which were tested, only TIG-3 showed the observed suppression following LDRR? If yes, then the claim of differential response in normal versus cancer cells after LDRR is not experimentally justified. Authors should include the data for all tested cell lines (both normal and cancer) and if only one cell line shows the effect among many normal cell lines tested then should avoid generalizing the findings. A549 did not show any changes in proliferation but what about the other cancer cell lines tested? Authors should clarify these issues and include the relevant data with all cell lines tested.

Response 1: We did not include the results of radio-sensitivity comparison among several cell lines in the original submission because the experimental conditions were different from those in Figure 1. Based on the reviewer’s suggestion, we added the results of radio-sensitivity comparison among several cell lines as Supplementary Figure S2. Those results suggest that normal cells are more sensitive to LDR-IR than cancer cells. We used two normal cell lines, TIG-3 (fibroblast from lung) and KD (fibroblast form lip), in those experiments, and we found that KD was also sensitive to radiation but did not grow well enough for use in the experiments. We thus chose TIG-3 cells for all experiments throughout the study. Nevertheless, we agree with the reviewer and added sentences in the Results and Discussion sections which indicate that our results are one example but not generalizable without limitations (Lanes 361-365).

2: The irradiation procedure is the key to this kind of experiment. The irradiation procedure should be presented as an independent section under Materials and Methods (M&M) and should be sufficiently elaborated to help the readers understand the exact procedure for irradiation. The authors should mention clearly which dose was delivered on the indicated dose rate? If the dose was 100, 500, or 1000mGy then does it mean that it took a whole day to irradiate the sample! Does per day mean 12 or 24 hours? Did some experiments take 1-7 days for irradiation (as mentioned in the M&M section)? If yes how the temperature and CO2 were maintained during irradiation?

Response 2: We thank the reviewer for pointing out our terse explanation of the experimental procedure for LDR-IR. We added a section explaining the irradiation procedure in the Materials and Methods section. (Lines 83-89)

3: Figure 1: Why MTT assay was preferred over the clonogenic survival assay? MTT only shows the impact on short-term proliferation. The addition of clonogenic survival data is necessary to access the effect of LDRR properly.

Response 3: As the reviewer points out, the clonogenic survival assay is often used to investigate the effects of radiation over long-term proliferation. Since we found cell-type differences of radio-sensitivity in preliminary experiments within 4 days of exposure, we chose the MTT assay to evaluate short-term proliferation. Recently, the MTT assay and other methods, including the standard clonogenic survival assay, have been employed to evaluate cellular sensitivity to radiation in numerous reports. We tried to clarify the intracellular signal transduction caused by LDR-IR in the short term, but it would be interesting to observe it in the long term as a next step. We would therefore like to employ the clonogenic survival assay in our next study.

4: Figure 2: Why the gene expression was not checked in A549 cells! Although no effect of LDR-IR was observed on the proliferation of A549, still it is important and valid to compare gene expression analysis between normal vs cancer cells.

Response 4: We definitely agree with the reviewer: it would be preferable to compare several cell lines in the RNA-seq analysis. Unfortunately, we could only subject TIG-3 cells to comprehensive gene expression analyses because of a limited budget. For the same reason we compared expression levels of only a limited number of genes by quantitative RT-PCR analyses.

5: Figure 5a: LDRR exerts similar effects on the cell cycle in TIG-3 and A549 cells which suggests that both normal and cancer cells have a similar impact of LDRR! This result contradicts the claim of the cell type-based response of LDRR.

Response 5: We had a similar concern. In the cell proliferation assay, we observed a maximum of 20% reduction of TIG-3 cells. On the other hand, we found that although only a small proportion of cells was modified in the analyses of cell cycle and apoptotic signal, it was statistically significant. These findings suggest that other mechanisms, e.g. ferroptosis or autophagy, underlie the diversity in radio-sensitivity. It would be important to examine other pathways, such as cell death or cell-cycle arrest, as well as molecular signals such as p53, which are known to play an important role in radiation response. One report revealed mutations in molecules downstream of p53 in A549 cells, although TIG-3 and A549 cells have wild-type p53. We added sentences related to this issue in the Discussion.

6: Figure 5b: It is surprising that TIG-3 and A549 show similar cell cycle responses in Figure 1A but clear differences on M-phase cells. Authors should check the mitotic cells using H3ps10 staining combined with cell cycle distribution to strengthen their results claiming differences in G2-M transition between these two cell lines. In addition, the representative images of mitotic cells observed under the microscope should be included.

Response 6: We understand that flow cytometry analysis using H3ps10 staining could strengthen our results. However, because we did not do that experiment, it would take time to confirm the results that way. Instead, we showed that checkpoint control might differ by counting cells in the M phase.

As the reviewer suggested, we added representative photos of M-phase cells in the Supplementary Materials (Fig S6).

7: Figure 6: This is an important observation and this response should also be checked in several cell lines which authors already checked for proliferation during the screening phase (please also see comment 1). Inclusion of such data with clear differences among many normal and cancer cell lines would strengthen the findings significantly.

Response 7: Because the results in Figure 6 are quite interesting, as the reviewer pointed out, it would be important to compare effects of LDR-IR on chemo-sensitivity among various cell lines. We added results of those experiments in HSC2 cells in the Supplementary Materials (Fig S7). For the same reason as in Response 1, we used TIG-3 cells as a normal cell line. We therefore also added sentences about our budget limitation in the Discussion (Lanes361-365).

8: The supplementary data was not accessible using the given link and showed the Error 404-File not found!

Response 8: This is unfortunate and we are sorry for the reviewer’s trouble. We have asked the editorial office to correct this problem so that reviewers and readers will be able to access the Supplementary Materials.

Round 2

Reviewer 2 Report

The authors have adequately addressed my concerns and the revised version has improved significantly. The paper is suitable for publication in Cells.